# IMPROVE WEAKLY SUPERVISED VISUAL GROUNDING BY LEARNING WHERE TO FOCUS ON

## ABSTRACT

Visual grounding is a crucial task for connecting visual and language descriptions by identifying target objects based on language entities. However, fully supervised methods require extensive annotations, which can be challenging and time-consuming to obtain. Weakly supervised visual grounding, which only relies on image-sentence association without object-level annotations, offers a promising solution. Previous approaches have mainly focused on finding the relationship between detected candidates, without considering improving object localization. In this work, we propose a novel method that leverages Grad-CAM to help the model identify precise objects. Specifically, we introduce a CAM encoder that exploits Grad-CAM information and a new loss function, attention mining loss, to guide the Grad-CAM feature to focus on the entire object. We also use an architecture which combines CNN and transformer, and a multi-modality fusion module to aggregate visual features, language features and CAM features. Our proposed approach achieves state-of-the-art results on several datasets, demonstrating its effectiveness in different scenes. Ablation studies further confirm the benefits of our architecture.

## 1 INTRODUCTION

As computer vision and natural language processing continue to advance, cross-modality understanding has become increasingly important in deep learning and has attracted significant attention. Many tasks aim to connect visual understanding and language description, including visual commonsense reasoning (Zellers et al. (2019)), visual question answering (Peng et al. (2019)), visual captioning (Kuo & Kira (2022)), and visual grounding (Liu et al. (2021)). Visual grounding is particularly crucial in these tasks because it involves identifying relevant visual regions based on language entities. This requires the model to localize objects accurately and comprehend the semantic-level information in the given language query. Thus, visual grounding can serve as a critical component in other vision-language tasks.

Recent visual grounding methods can be classified into two categories: fully supervised visual grounding and weakly supervised visual grounding. Although fully supervised visual grounding achieves impressive performance, it relies on manual annotations, which can be time-consuming. In contrast, weakly supervised visual grounding only requires corresponding images and language descriptions without region-level annotations, making data collection easier. However, weakly supervised visual grounding faces several challenges due to the lack of information from annotations, such as localizing different objects in one image, understanding the relationship between different entities, and comprehending the attributes of every entity.

Previous weakly supervised visual grounding methods mostly focus on detecting multiple entities using a pre-trained object detector and finding relationships between them. In contrast, our approach aims to enhance the performance of weakly supervised visual grounding by improving object localization and enabling the model to better focus on the target objects. To achieve this, we propose a method that combines Grad-CAM with weakly supervised visual grounding methods. Specifically, we design a CAM encoder to extract information from the Grad-CAM and introduce an attention mining loss to force the Grad-CAM features to focus on the whole object rather than just part of it. The loss function generates interested and uninterested regions based on the Grad-CAM feature, and the CNN backbone provides a confidence score for the uninterested regions. If the score is high,

Figure 1: Grad-CAM is frequently utilized in weakly supervised training scenarios. Nevertheless, no previous attempts have been made to integrate Grad-CAM with existing weakly supervised visual grounding methods. With this in mind, we introduced a technique that utilizes Grad-CAM to boost the performance of weakly supervised visual grounding. The results depicted in the figure show that without the use of Grad-CAM, the predicted region for the discovered dog is not very accurate. In contrast, the predictions are significantly more precise when Grad-CAM is integrated.

it indicates that parts of some objects are still in the uninterested regions, and we penalize it using a loss. By using this loss function, our Grad-CAM can better focus on the target objects and improve object localization.

Our proposed architecture includes the CAM encoder and another architecture that combines the transformer and CNN in the visual encoder. The traditional transformer-based methods can be challenging to train due to their large scale, whereas our proposed architecture can make the training process more efficient. Our visual encoder provides multi-layer features that offer both high-level and low-level information about the original image. For the language encoder, we adopt an architecture similar to TransVG (Deng et al. (2021)), which follows the original BERT (Devlin et al. (2018)) architecture. This enables our model to better understand the input language query by processing the entire sentence, rather than just the word embeddings.

To combine the information from the visual features, language features, and CAM features, we introduce a multi-modality fusion module that can aggregate the multi-modality information and make use of the multi-level visual features. Then, we produce the final bounding box prediction using the regression prediction head.

To evaluate our proposed methods, we performed extensive experiments on five different datasets and compared them with the latest weakly supervised visual grounding methods as well as fully supervised visual grounding methods. Fully supervised visual grounding methods represent the upper bound of the visual grounding tasks. Our experimental results demonstrate that our methods achieve state-of-the-art results on different datasets. We also conducted numerous ablation studies to demonstrate the effectiveness of our proposed CAM encoder and other module designs. Furthermore, we provided many visualization results to prove our architecture's capability in dealing with various scenes.

In summary, our contributions are as follows:

- We proposed the use of Grad-CAM to improve the localization of weakly supervised visual grounding. Specifically, we designed a CAM encoder to extract information from the Grad-CAM features and a loss function called attention mining loss that forces the Grad-CAM features to focus on the whole object rather than just parts of it.

- We proposed a module that combines the CNN architecture and the transformer architecture as the visual encoder. Additionally, we provided multi-layer features in our visual encoder to provide low-level coarse information and high-level semantic representations. We made use of multi-layer information in our multi-modality fusion module.

- We performed extensive experiments on five datasets and achieved state-of-the-art results in four datasets. We also conducted many ablation studies to demonstrate the effectiveness of our proposed modules and loss function and presented numerous visualizations to demonstrate our model's ability to deal with various scenes.

## 2 RELATED WORKS

### 2.1 VISUAL GROUNDING

Visual grounding refers to the task of finding the most relevant region in an image that corresponds to a given natural language query. Several datasets such as ReferItGame (Kazemzadeh et al. (2014)), Flickr30k (Plummer et al. (2015)), RefCOCO (Yu et al. (2016)), and Visual Genome (Mao et al. (2016)) have been used to evaluate various methods. Some methods calculate the similarity between the candidate region and the language embedding (Plummer et al. (2018)). In addition, some work such as TransVG (Deng et al. (2021)) and ViLG (Du et al. (2022)) have explored the use of transformer-based architectures to solve visual grounding tasks. Others have attempted to combine visual grounding with other tasks, such as image caption alignment (Datta et al. (2019)), and audio-grounding (Wang et al. (2022)). Recent works have extended 2D image visual grounding to 3D video visual grounding, such as in Multi-Stream VRG (Huang et al. (2022)) and LanguageRefer (Roh et al. (2022)). These fully supervised visual grounding methods have achieved impressive results. However, collecting annotations for these datasets is a time-consuming and expensive process. So, we try to propose a weakly supervised visual grounding method that relies on easier collected data.

### 2.2 WEAKLY SUPERVISED VISUAL GROUNDING

While fully supervised visual grounding requires time-consuming and expensive annotations, weakly supervised visual grounding only needs aligned sentences and images without region-sentence correspondence. Weakly supervised training is commonly used in other areas (Li et al. (2021); Meng et al. (2018)). There have been many efforts to improve the performance of weakly supervised visual grounding, such as GVD-CVAE (Mavroudi & Vidal (2022)), which uses a conditional generative model to learn its approximate posterior distribution given the full sentence, RIF (Liu et al. (2021)) which learns the relationship between different detected candidates to improve task performance, and Pseudo-Q (Jiang et al. (2022)) which generates a pseudo phrase to provide more information in the language embedding to improve performance. Some researchers have also extended weakly supervised visual grounding from 2D image to 3D video (Yang et al. (2020a); Shi et al. (2019)). Although many methods have been proposed to improve the performance of weakly supervised visual grounding, none have made use of Grad-CAM, which is commonly used in weakly supervised training. Therefore, in this paper, we propose a method to use Grad-CAM to improve the performance of weakly supervised visual grounding.

### 2.3 VISION-LANGUAGE MODEL

The vision-language model has gained significant attention in recent years due to advancements in computer vision and natural language processing. This model has various applications, including VQA (Peng et al. (2019)), image captioning (Kuo & Kira (2022)), and commonsense reasoning (Zellers et al. (2019)). Most of the existing vision-language models rely on large pre-trained models. However, recent efforts (Deng et al. (2021)) have focused on using the vision-language transformer to solve these problems, inspired by ViT (Dosovitskiy et al. (2020)) and DETR (Carion et al. (2020)). Despite its potential, the vision-language transformer's large scale makes it challenging to train. To address this issue, we propose a method that combines the transformer architecture and CNN architecture in the visual encoder, enabling us to speed up the training process.

## 3 METHODS

**Architecture** Our proposed CAM-based weakly supervised visual grounding architecture has four important modules: visual encoder, language encoder, CAM encoder, and a multi-modality fusion module. The whole architecture can be seen in Fig2

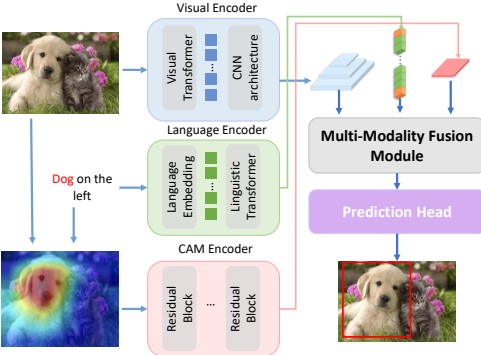

Figure 2: The architecture of our model is composed of four essential components: a visual encoder that receives the original image as input and outputs a visual embedding, a language encoder that receives the language query as input and outputs a language embedding, a CAM encoder that takes the Grad-CAM representation as input and outputs a CAM embedding, and a multi-modality fusion module that aims to combine the information from the visual embedding, language embedding, and CAM embedding.

## 3.1 VISUAL ENCODER

The visual encoder is a crucial component of our model as it is responsible for providing an appropriate representation of the input image. To achieve this, we have designed an architecture for the visual encoder, which can be seen in Figure.3. Our approach combines the transformer and CNN architectures in the visual encoder to reduce computation costs and accelerate the training process.

To begin with, given an input image $I \in \mathbb{R}^{H \times W \times 3}$, we first use a pre-trained detector that has been trained on the Visual Genome dataset (Krishna et al. (2017)) to detect candidate regions in the input image. For each candidate and the original image, we then use a pre-trained CNN network (in our case, the ResNet50 (He et al. (2016))) to obtain the intermediate feature map of the input image. We use the last three layers' features $F_3 - F_5$ as our intermediate feature map.

Inspired by DETR (Carion et al. (2020)), which provides a position embedding to the original feature map, we also calculate the position embedding for the intermediate features and add it to the intermediate features. This results in three new features $F_3^{(1)} - F_5^{(1)}$, which now incorporate positional information.

Next, we use a self-attention module to capture the local information of the given features, which we call $F_3^{(2)} - F_5^{(2)}$. By using position encoding, the transformer architecture can better understand the relationship between different tokens. We then add $F_3^{(1)} - F_5^{(1)}$ and $F_3^{(2)} - F_5^{(2)}$ and pass the result to a CNN module to speed up the training process. The CNN module consists of a $3 \times 3$ Convolution layer, group normalization, and GELU activation. After the GELU activation, we add another $3 \times 3$ convolution layer to obtain the final features $F_3^{(3)} - F_5^{(3)}$.

Finally, we input the result of the visual encoder into our multi-modality fusion module. Overall, our visual encoder architecture effectively combines the strengths of both the transformer and CNN models to provide a robust and efficient representation of the input image, which can significantly enhance the performance of our weakly supervised visual grounding model.

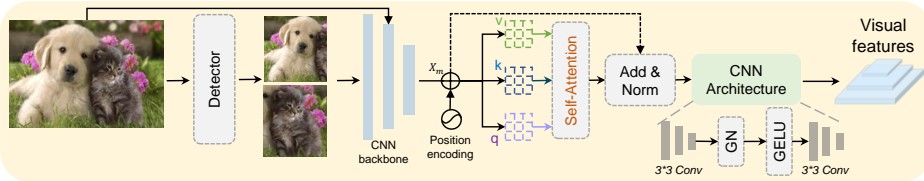

Figure 3: Architecture of our visual encoder. Our visual encoder is designed to obtain visual features from input images. First, we use a pre-trained detector to identify several candidates within the input image. Then, we extract features using a CNN backbone for each candidate and the original input image. After obtaining the CNN backbone features, we add position embeddings and apply a self-attention module to extract local information. Then, a CNN architecture is used to obtain the final visual features.

Figure 4: **Left**: Architecture of our language encoder. Our language encoder adopts a similar architecture as TransVG. Given an input language query, we first obtain the word embedding for each word in the sentence. Then, we add a [CLS] token at the beginning and an [SEP] token at the end of the sentence. Subsequently, we employ a linguistic transformer to compute the final language features, which capture semantic-level information instead of word-level information. **Middle**: Our proposed CAM encoder is designed to extract features using Grad-CAM based on an input image and a language query. To achieve this, we first identify the noun in the query and then use Grad-CAM to extract its features. Next, we use several residual blocks on the CNN backbone feature to obtain the final CAM features. Additionally, we introduce an attention-mining loss to enhance the overall performance of our model. **Right**: Our proposed CAM encoder is designed to extract features using Grad-CAM based on an input image and a language query. We first identify the noun in the query and then use Grad-CAM to extract its features to achieve this. Next, we use several residual blocks on the CNN backbone feature to obtain the final CAM features. Additionally, we introduce an attention-mining loss to enhance the overall performance of our model.

## 3.2 LANGUAGE ENCODER

The language encoder serves the purpose of generating an embedding for the input sentence. In Fig4, we present the detailed architecture of our language encoder. To leverage the pre-trained BERT (Devlin et al. (2018)) model, we adopt a similar approach to TransVG (Deng et al. (2021)) in designing the language encoder. We first convert each word into a one-hot vector for a given input sentence and obtain its corresponding embedding by referring to the token table. We then add a [CLS] token at the beginning and a [SEP] token at the end of the tokenized sentence. The tokenized sentence is then passed through a linguistic transformer that has a structure similar to the basic BERT model, comprising of 12 transformer encoder layers. This allows us to obtain a language embedding that captures semantic-level information for the entire sentence rather than just word-level information. We use the resulting language features $F_l \in \mathbb{R}^{C_l \times N_l}$, where $C_l$ represents the output dimension of the transformer encoder and $N_l$ is the number of language tokens, as an input to the multi-modality fusion module.

## 3.3 CAM ENCODER

The CAM encoder is an important component of our model, which is designed to utilize the results of the Grad-CAM method (Selvaraju et al. (2017)) and provide a module that can extract the relevant information from the Grad-CAM attention map. The detailed architecture of the CAM encoder is illustrated in Fig 4. The primary objective of this module is to help the model obtain a better predicted bounding box by obtaining the necessary information from the Grad-CAM attention map.

We first identify the nouns in the input language query to accomplish this task, as depicted in Fig 4. For instance, given an input language query "Dog on the left", we will first identify "dog" in the sentence, which is then treated as the label, and the Grad-CAM is used to obtain the attention map for the dog. If there are multiple nouns in the sentence, we will obtain the attention map for each noun separately. However, the original Grad-CAM method can only identify the target object and cannot cover all objects in the image. Therefore, we designed a module that can extract the information from the Grad-CAM and identify the whole object.

After obtaining the Grad-CAM result, we use several residual blocks that follow the same structure as the ResNet50 (He et al. (2016)) to extract the information from the Grad-CAM feature. Subsequently, we obtain the final CAM features. Our goal is to force the final CAM features to focus on the whole object, rather than just a part of it. To achieve this objective, we introduce a new loss function, called attention mining loss, which is inspired by GAIN (Li et al. (2018)). This loss function can help the final Grad-CAM features to focus on the whole object. The formulation of the attention mining loss is $L_{am} = S^c(I*)$, where $S^c$ is the prediction score for class $c$ for the CNN

backbone, and $I*$ is the uninterested region of the final CAM features. Specifically, we set a threshold for the final Grad-CAM feature and find the interested region $I$ using the Grad-CAM features. The remaining region is taken as the uninterested region. The objective of our attention mining loss is to ensure that if the final CAM features do not focus on the whole object, the uninterested region will contain a part of the object, and the CNN backbone will give a confidence score that is not zero for that class. Then, our attention mining loss will force the module to contain the interested region as much as possible. Using the designed attention mining loss, we force the CAM-feature focus on the object and can provide the location for every object in the original image.

### 3.4 MULTI-MODALITY FUSION MODULE

In our approach, we leverage the benefits of multi-modal information by fusing visual, language, and class activation map (CAM) embeddings together. However, in order to effectively utilize these different modalities, we need a module that can aggregate the information of these embeddings.

To address this, we introduce our multi-modality fusion module. The detailed architecture can be seen in Fig4. In this module, the input visual feature and CAM feature are not queries. Hence, we first grid and flatten them. Notice that the input visual features contain multi-layer information because we use multi-level features of the CNN backbone in the visual encoder. The advantage of using multi-level features is that they can provide different information, whereas low-level features can provide more coarse information like the shape and edge of the object, and high-level features can provide more semantic information like the attribute and the class of the object. Here, we can get three different visual queries that contain different information.

After flattening, the dimension of the visual tokens, CAM tokens, and linguistic tokens is different, hence, we add two linear projection layers (one for each modality and one for the attention layer) to project every token to the same dimension $C_p$. This ensures that all tokens are in the same space.

In order to make use of the multi-level information, we use three different attention layers to aggregate information. For these three attention layers, we will input all of the language tokens and the CAM tokens. For the visual tokens, we will only input one layer of the query tokens to every attention layer. This is because we want each attention layer to focus on different aspects of the visual features. We can extract different levels of information by inputting a different visual query to each attention layer. The input of every attention layer can be written as follows:

$$x_i = [\overbrace{P_v^{i1}, \cdots p_v^{iN_{vi}}}^{\text{visual tokens } p_v^i}, \quad \underbrace{p_l^1, \cdots p_l^{N_l}}_{\text{linguistic tokens } p_l}, \quad \overbrace{p_c^1, \cdots p_c^{N_c}}^{\text{CAM tokens } p_c}, P_r] \tag{1}$$

Here, $x_i$ means the input for the $i_{th}$ attention layer, $p_v^{in}$ means the $n_{th}$ visual tokens of the $i_{th}$ feature layer, $p_l^n$ means the $n_{th}$ linguistic tokens, and $p_c^n$ means the $n_{th}$ CAM tokens. $N_{vi}$ means the total number of visual tokens of the $i_{th}$ features layers, $N_l$ means the total number of linguistic tokens, and $N_c$ means the total number of the CAM tokens. Following TransVG (Deng et al. (2021)), we pre-defined a learnable embedding [REG] token $P_r$ at the end of the input.

We hope every attention layer can extract semantic information through this design. After the attention layer, we directly concatenate the output of the attention layers. Then, we use a regression prediction head composed of three fully connected layers to calculate the final bounding box.

Overall, the multi-modality fusion module is designed to effectively aggregate information from different modalities and leverage the benefits of multi-level visual features to improve object detection.

### 3.5 LOSS FUNCTION

For the loss function, except for the attention mining loss $L_{am}$, we also use a self-taught regression loss $L_{reg}$ and a phrase reconstruction loss $L_{rec}$ following RIF (Liu et al. (2021)). Hence the final formulation for the loss function can be written as

$$L = \lambda_{reg}L_{reg} + \lambda_{rec}L_{rec} + \lambda_{am}L_{am} \tag{2}$$

In our experiments, we set the $\lambda_{reg}$ to 0.1, $\lambda_{rec}$ and the $\lambda_{am}$ to 1.

# 4 EXPERIMENTS

## 4.1 EXPERIMENTS SETUP

### 4.1.1 DATASET

We conducted experiments on five different datasets: RefCOCO (Yu et al. (2016)), RefCOCO+ (Yu et al. (2016)), RefCOCOg (Mao et al. (2016)), ReferItGame (Kazemzadeh et al. (2014)), and Flickr30K Entities (Plummer et al. (2015)). The train and test splits were consistent with the (Deng et al. (2021)) setting. The training set for these datasets contained a total of 16,994, 16,992, 24,698, 8,994, and 29,779 instances, respectively.

### 4.1.2 IMPLEMENT DETAILS

We used a pre-trained detector trained on the Visual Genome dataset (Krishna et al. (2017)) containing 1,600 objects. For word embedding, we utilized a pre-trained BERT (Devlin et al. (2018)). To obtain Grad-CAM features, we used a pre-trained ResNet (He et al. (2016)) trained on ImageNet (Russakovsky et al. (2015)). In the multi-modality fusion module, the dimensions for $P_v^i \in \mathbb{R}^{C_p \times N_{vi}}$, $P_l \in \mathbb{R}^{C_p \times N_l}$, $P_c \in \mathbb{R}^{C_p \times N_c}$, and the [REG] token $P_r \in \mathbb{R}^{C_p \times 1}$ were set to 256 in our experiments. The [REG] token was randomly initialized at the start of training and optimized during the training process.

## 4.2 COMPARISON WITH THE STATE-OF-THE-ARTS

We present the quantitative results of our proposed methods compared to the latest approaches. To provide a detailed comparison, we evaluate not only weakly supervised visual grounding but also fully supervised methods. Our table reports the Top-1 accuracy results, where a predicted bounding box is considered correct if its Jaccard overlap with the ground truth is greater than 0.5, otherwise, it is treated as false.

**RefCOCO/RefCOCO+/RefCOCOg**: Table1 shows the accuracy of our methods on RefCOCO, RefCOCO+, and RefCOCOg datasets. Our model surpasses the current state-of-the-art in Ref-COCO+ and RefCOCOg and achieves very close performance to the current state-of-the-art in RefCOCO. It is worth noting that Pseudo-Q has a significant gap in the testA and testB split of RefCOCO+, and its poor performance on testB split affects its final results in RefCOCO+. In contrast, our methods have tiny gaps between the testA and testB splits and outperform the current state-of-the-art in all cases.

**ReferItGame**: Table2 reports the accuracy of our methods. Our model achieves 45.27% top-1 accuracy, outperforming previous methods by 1.95%.

Table 1: Quantitative results of our proposed weakly supervised visual grounding model in RefCOCO, RefCOCO+, and RefCOCOg, the best results are highlighted in **bold** and the second best results are highlighted in underlined. Sup. means the supervision level. Here we provide multiple latest weakly supervised and fully supervised visual grounding results

| method | Sup. | RefCOCO | | | RefCOCO+ | | | RefCOCOg | | |
|---|---|---|---|---|---|---|---|---|---|---|
| | | val | testA | testB | val | testA | testB | val-g | val-u | test-u |
| VC (Zhang et al. (2018)) | Weak | - | 33.29 | 30.13 | - | 34.60 | 31.58 | 33.79 | - | - |
| ARN (Liu et al. (2019b)) | | 34.26 | 36.43 | 33.07 | 34.53 | 36.01 | 33.75 | 33.75 | - | - |
| KPRN (Liu et al. (2019c)) | | 35.04 | 34.74 | 36.98 | 35.96 | 35.24 | 36.96 | 33.56 | - | - |
| DTWREG (Sun et al. (2021)) | | 39.21 | 41.14 | 37.72 | 39.18 | 40.10 | 38.08 | 43.24 | - | - |
| Pseudo-Q Jiang et al. (2022) | | 56.02 | 58.25 | 54.13 | 38.88 | 45.06 | 32.13 | 49.82 | 46.25 | 47.44 |
| Ours | | 54.78 | 55.71 | 53.15 | 42.25 | 45.88 | 39.18 | 50.48 | 48.54 | 51.25 |
| MAttNet (Yu et al. (2018a)) | Full | 76.65 | 81.14 | 69.99 | 65.33 | 71.62 | 56.02 | - | 66.58 | 67.27 |
| NMTree (Liu et al. (2019a)) | | 76.41 | 81.21 | 70.09 | 66.46 | 72.02 | 57.52 | 64.62 | 65.87 | 66.44 |
| FAOA (Yang et al. (2019)) | | 72.54 | 74.35 | 68.50 | 56.81 | 60.23 | 49.60 | 56.12 | 61.33 | 60.36 |
| ReSC (Yang et al. (2020b)) | | 77.63 | 80.45 | 72.30 | 63.59 | 68.36 | 56.81 | 63.12 | 67.30 | 67.20 |
| TransVG (Deng et al. (2021)) | | 80.32 | 82.67 | 78.12 | 63.50 | 68.15 | 55.63 | 66.56 | 67.66 | 67.44 |

Table 2: Quantitative results of our proposed weakly supervised visual grounding model in ReferItGame, Flicker30K+, the best results are highlighted in **bold** and the second best results are highlighted in underlined.

| method | Sup. | ReferItGame | Flicker30K |
|---|---|---|---|
| KACNet (Chen et al. (2018)) | Weak | 33.67 | 46.61 |
| MATN (Zhao et al. (2018)) | | 33.10 | 13.61 |
| ARN (Liu et al. (2019b)) | | 26.19 | - |
| CLWS (Gupta et al. (2020)) | | - | 51.67 |
| RIF (Liu et al. (2021)) | | 37.68 | 59.27 |
| CKD (Wang et al. (2021)) | | 38.39 | 53.10 |
| Pseudo-Q Jiang et al. (2022) | | 43.32 | 60.41 |
| Ours | | 45.27 | 61.78 |
| PircNet (Kovvuri & Nevatia (2019)) | Full | 59.13 | 72.93 |
| DDPN (Yu et al. (2018b)) | | 63.00 | 73.30 |
| FAOA (Yang et al. (2019)) | | 60.67 | 68.71 |
| ReSC (Yang et al. (2020b)) | | 64.60 | 69.28 |
| TransVG (Deng et al. (2021)) | | 69.76 | 78.47 |

Table 3: Ablation for the CAM Encoder, the results show that our proposed CAM encoder and attention mining loss can obviously improve the performance of the model.

| method | RefCOCO | | | RefCOCO+ | | | RefCOCOg | | | ReferItGame | Flicker30K |
|---|---|---|---|---|---|---|---|---|---|---|---|
| | val | testA | testB | val | testA | testB | val-g | val-u | test-u | | |
| w/o CAM encoder, w/o AM loss | 49.38 | 50.18 | 47.96 | 37.58 | 39.81 | 35.08 | 44.97 | 43.09 | 45.05 | 37.98 | 57.78 |
| w/ CAM encoder,w/o AM loss | 52.47 | 53.36 | 51.19 | 41.02 | 44.09 | 38.91 | 47.68 | 46.81 | 49.18 | 42.68 | 58.87 |
| w/ CAM encoder, w/ AM loss | 54.78 | 55.71 | 53.15 | 42.25 | 45.88 | 39.18 | 50.48 | 48.54 | 51.25 | 45.27 | 61.78 |

Table 4: Ablation for the Multi-layer features, the results show that providing multi-layer features can help the model capture high-level and low-level information which can obviously improve the performance of the model

| features | RefCOCO | | | RefCOCO+ | | | RefCOCOg | | | ReferItGame | Flicker30K |
|---|---|---|---|---|---|---|---|---|---|---|---|
| | val | testA | testB | val | testA | testB | val-g | val-u | test-u | | |
| $F_3$ | 49.81 | 51.39 | 48.27 | 38.01 | 41.97 | 35.13 | 46.19 | 44.84 | 47.77 | 42.17 | 57.38 |
| $F_4$ | 49.95 | 51.36 | 47.93 | 37.98 | 40.33 | 34.86 | 45.51 | 44.67 | 47.16 | 40.07 | 56.75 |
| $F_5$ | 50.18 | 51.17 | 48.96 | 37.95 | 40.19 | 34.91 | 45.91 | 44.69 | 46.89 | 40.58 | 57.19 |
| $F_3.F_4$ | 52.35 | 54.29 | 50.31 | 39.65 | 42.39 | 37.99 | 47.18 | 46.87 | 48.92 | 42.95 | 57.92 |
| $F_3, F_5$ | 53.38 | 55.16 | 51.53 | 41.92 | 43.36 | 38.75 | 49.68 | 47.36 | 51.09 | 44.81 | 59.91 |
| $F_4, F_5$ | 53.85 | 55.05 | 52.49 | 41.94 | 45.48 | 38.39 | 49.46 | 47.91 | 51.13 | 44.62 | 59.71 |
| $F_3, F_4, F_5$ | 54.78 | 55.71 | 53.15 | 42.25 | 45.88 | 39.18 | 50.48 | 48.54 | 51.25 | 45.27 | 61.78 |

**Flickr30K**: Table2 shows the accuracy of our methods. Our model achieves 67.78% top-1 accuracy, outperforming previous methods by 1.37%.

## 4.3 ABLATION STUDIES

### 4.3.1 EFFECTIVENESS OF THE CAM ENCODER

In this section, we conduct an ablation study to demonstrate the effectiveness of our proposed CAM encoder and attention-mining loss. Table3 shows the results for different settings, where "w/" and "w/o" indicate whether the CAM encoder and attention mining loss are included in our architecture during training. We observe that the proposed CAM encoder contributes to a 5% increase in top-1 accuracy compared to the baseline model. Furthermore, the attention mining loss improves the accuracy by 2%. These results demonstrate the effectiveness of our proposed CAM encoder.

### 4.3.2 EFFECTIVENESS OF MULTI-LAYER FEATURES

In this section, we provide an ablation study to demonstrate the effectiveness of multi-layer features in the visual encoder. Table4 shows the quantitative results for five datasets using one layer feature and two different layer features. We observe that using only one layer feature leads to a drop in performance of about 4%, and even using two different layer features still leads to a drop of about 2%. This ablation study highlights the effectiveness of using multi-layer features in the visual encoder.

### 4.3.3 EFFECTIVENESS OF THE MULTI-MODALITY FUSION MODULE

In this section, we conduct an ablation study to evaluate the effectiveness of our multi-modality fusion module. Table5 shows the results for three different settings. The "one attention layer" setting uses a single attention layer to process all visual, language, and CAM features, followed by the prediction head. The "three attention layers, $1 \times 1$ conv" setting uses three different attention layers but aggregates the output of the attention layers using a $1 \times 1$ convolution layer. We observe that both settings slightly decrease the performance of our model. Therefore, our main paper uses three different attention layers and concatenates their output to predict the final bounding box.

Table 5: Ablation for the multi-modality fusion module, the results show that multi-level features that contain coarse information and semantic information are helpful for the weakly supervised visual grounding. Also, concatenate operations are better than $1 \times 1$ convolution in our model.

| method | RefCOCO | | | RefCOCO+ | | | RefCOCOg | | | ReferItGame | Flicker30K |
|---|---|---|---|---|---|---|---|---|---|---|---|
| | val | testA | testB | val | testA | testB | val-g | val-u | test-u | | |
| one attention layer | 52.97 | 54.32 | 50.77 | 41.10 | 44.13 | 38.79 | 47.77 | 46.23 | 49.13 | 43.53 | 59.02 |
| three attention layer, $1 \times 1$ conv | 53.15 | 54.35 | 51.93 | 42.11 | 45.33 | 39.07 | 49.91 | 48.97 | 51.03 | 44.97 | 61.57 |
| three attention layer, concatenate | 54.78 | 55.71 | 53.15 | 42.25 | 45.88 | 39.18 | 50.48 | 48.54 | 51.25 | 45.27 | 61.78 |

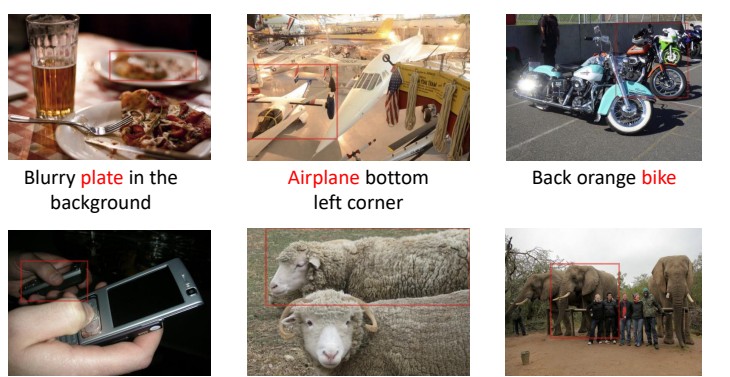

Blurry plate in the background

Airplane bottom left corner

Back orange bike

Bed on far left

Small device in left hand

Sheep farthest from the screen

Second from left elephant

Teddy bear left

Figure 5: The visualization demonstrates the effectiveness of our proposed architecture. For every language query, we use the red color to highlight the nouns. From the visualization, we can see that our model can have impressive performance even when the target object is blurred or there are multiple similar objects

## 4.4 VISUALIZATIONS

This section provides visualizations of the proposed model to demonstrate its effectiveness. As shown in Figure 6, the proposed model performs well in various scenes. It can identify the target object even when it is not very clear, as in the case of the plate, and distinguish it from similar objects, such as the sheep and the elephant. This indicates that the model can comprehend the spatial information in the given language query and align it with the image candidate. Moreover, the model can understand the meaning of color, as demonstrated in the example of the bike, where the model identifies the bike with the correct color. Even when the input language query includes multiple nouns, the model can still accurately identify the target object. These visualizations demonstrate the model's ability to handle different scenes comprehend the semantic-level information in the input language query and align it with the different candidates in the given image.

## 5 CONCLUSION

In conclusion, we propose a novel weakly supervised visual grounding architecture that combines the transformer and CNN architectures. Observing that Grad-CAM is useful in weakly supervised training, we design a CAM encoder that utilizes the Grad-CAM to provide better object localization when predicting the final bounding box. However, the original Grad-CAM can only identify the target object and may not focus on the whole object. Therefore, we introduce a new attention-mining loss that forces the Grad-CAM to focus on the whole object instead of only a part of it. Besides the CAM encoder, our proposed visual encoder also utilizes the transformer and CNN architectures to extract the visual and language features. Extensive experiments demonstrate the effectiveness of our proposed architecture, which achieves state-of-the-art performance on several datasets. Our proposed architecture can be transferred to tasks such as visual question answering.

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

## 6 APPENDIX

## 7 MORE DETAILS ABOUT THE LOSS FUNCTION

In this section, we will introduce more details about the self-taught regression loss and phrase reconstruction loss which following the RIF (Liu et al. (2021))

## 7.1 SELF-TAUGHT REGRESSION LOSS

The purpose of the self-taught regression loss is that the location for each phrase is not annotated, we need a loss function to help the model to find a location for each phrase. Here, we will use confident proposals from partially-trained models to supervise the location refinement. Specifically, given phrase $q_i$, we denote $\delta^{c*} = \{\delta i, m^{c*}\}$ where $\{\delta i, m^{c*}\}$ is the offset between proposal $o_m$ and the most confident proposal if their overlaps are larger than a threshold, otherwise we called is $\{\delta i, m^c\}$. Then the loss function is

$$L_{reg} = \sum_{i=1}^{N}(L_{sm}(\{\delta i, m^{c*}\}, \{\delta i, m^c\})) \tag{3}$$

where $L_{sm}$ is the smooth-L1 loss.

## 7.2 PHRASE RECONSTRUCTION LOSS

Given a noun phrase, we use a phrase reconstruction loss to provide model supervision. To use phrase reconstruction loss, we will first calculate a visual representation $z_i^c$ for each phrase $q_i$, then we can calculate a sequence of word distribution $y_i^c$ as below

$$y_i^c = \text{LSTM}_{dec}([z_i^c, q_i]) \tag{4}$$

Then we use a standard sequence log loss $L_{log}$ to calculate the final loss function which can be written as

$$L_{rec} = \sum_{i=1}^{N}(L_{log}(y_i^c, q_i)) \tag{5}$$

# 8 MORE ABLATION STUDIES

## 8.1 INFLUENCE OF THE NUMBER OF TRANSFORMER ENCODER LAYERS

In this section, we conduct an ablation study to evaluate the influence of the number of transformer encoder layers in our visual and language encoders. Table.6 shows the results for different numbers of encoder layers. We observe that if the number of encoder layers is too small, it is insufficient to extract all of the information in the given image and language query. Once the number of layers is sufficient to extract the information, further increasing the number of encoder layers does not improve performance.

# 9 MORE DETAILS ABOUT THE ARCHITECTURE

In this section, we provide more details about our architecture

## 9.1 RESOLUTION ABOUT MULTI-LAYER FEATURES

We provide multi-layer features to our visual encoder, here, we provide the resolution for different layers' features

Table 6: Ablation for the number of encoder layers in the visual and language encoder. The results show that increasing the encoder layer when the layer is not deep can improve the performance, and if we continue to increase the number of the encoder layers, the performance is not further improved; the best results are highlighted in bold

| visual encoder | language encoder | RefCOCO | | | RefCOCO+ | | | RefCOCOg | | | ReferItGame | Flicker30K |
|---|---|---|---|---|---|---|---|---|---|---|---|---|
| | | val | testA | testB | val | testA | testB | val-g | val-u | test-u | | |
| 3 | 3 | 50.13 | 51.11 | 48.68 | 38.16 | 41.34 | 36.90 | 45.94 | 43.30 | 47.65 | 39.71 | 56.34 |
| 3 | 6 | 52.68 | 53.91 | 51.35 | 40.95 | 42.78 | 38.47 | 47.06 | 45.18 | 49.76 | 42.29 | 59.82 |
| 6 | 6 | 53.86 | 55.07 | 52.67 | 41.29 | 44.88 | 38.23 | 49.18 | 47.33 | 51.11 | 44.39 | 60.37 |
| 6 | 12 | **54.78** | 55.71 | **53.15** | 42.25 | 45.88 | **39.18** | **50.48** | 48.54 | **51.25** | 45.27 | **61.78** |
| 12 | 12 | 54.73 | **55.82** | 53.01 | 41.28 | 44.79 | 38.11 | 49.58 | **48.55** | 50.19 | **45.39** | 61.25 |
| 12 | 24 | 53.27 | 54.49 | 51.13 | **42.46** | **46.98** | 38.77 | 50.19 | 48.30 | 50.79 | 44.95 | 60.98 |

Table 7: Input and output shape of different layers' features

| Layer | Input Shape |
|---|---|
| Layer$_5$ | $\frac{H \times W}{32}$ |
| Layer$_4$ | $\frac{H \times W}{16}$ |
| Layer$_3$ | $\frac{H \times W}{8}$ |

## 10  TRAINING DETAILS

In our experiments, we utilized two NVIDIA Tesla V100-sxm2 GPUs, each with 32GB of memory, for a total of 64GB of memory. All of the modules are end-to-end trained. We used the AdamW optimizer to optimize our architecture with an initial learning rate of 1e-5 and a weight decay of 1e-5. We used a batch size of 64 for all experiments. We applied the cosine learning rate schedule for all datasets. For data augmentation, we followed the same procedure as TransVG (Deng et al. (2021)), which included RandomBrightness, RandomContrast, RandomSaturation, ColorJitter, RandomResizeCrop, and RandomHorizonFlip. We trained our model for a total of 10 epochs for RefCOCO, RefCOCO+, and RefCOCOg datasets, and for 20 epochs for the ReferItGame and Flickr30K Entities datasets. These training parameters were chosen through experimentation to ensure that our model was optimized for performance on each dataset.

## 11  MORE VISUALIZATIONS

Here, we provide more visualizations of our model in Fig. 6

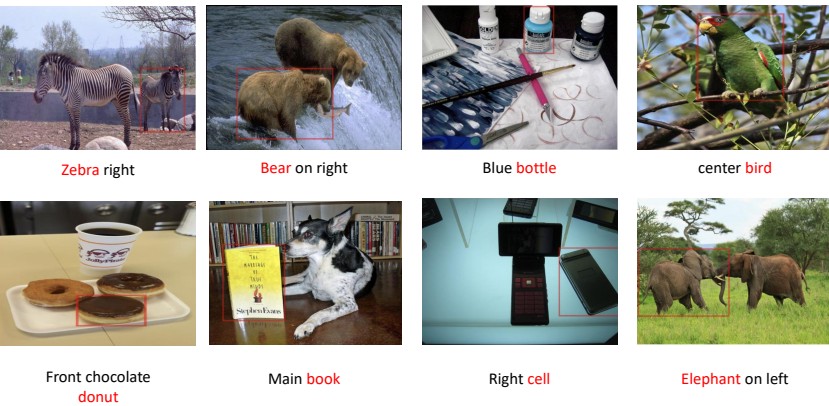

Figure 6: More visualizations of our model

