# OpenReview forum: "improve weakly supervised visual grounding by learning where to focus on"
_ICLR.cc/2025/Conference — Submitted to ICLR 2025_

### Official Review · Reviewer_2Uie · 2024-10-30

**Soundness:** 1
**Presentation:** 2
**Contribution:** 1
**Rating:** 3
**Confidence:** 5

**Summary:**

The paper focuses on the weakly-supervised learning setting in the Visual Grounding task. The authors propose to use Grad-CAM to explain the model's attention/focus and provide a loss to supervise the behavior of Grad-CAM. They also combine CNN and Transformer structures to provide vision features in different semantic levels.

**Strengths:**

1. Experiment has covered comprehensive datasets.
2. Achieve partial state-of-the-art performance on current benchmarks.
3. The writing of the idea is easy to understand.

**Weaknesses:**

1. **Motivation for the task is questionable**: In line 40 to line 42, the authors claim that Visual Grounding (VG) needs region-level annotation which is time consuming. Instead, weakly-supervised VG only needs the image-text pair, so it is easier to collect. However, the text in VG is still region-level annotation, like "a man in a white shirt next to the door". In human labeling scenario, I don't see that the additional bounding box (bbox) annotation occupies the main annotation effort given that the human already put an effort to come up with the detailed region-level sentence. **Therefore, this bbox annotation cost in VG needs a strict comparison to see if it matters or not.** In automatic labeling scenario, we can still use SOTA object detector to come up with bbox, and then use set-of-mask or any other region-level captioning technique to provide pseudo labels. Although there might be error for this automatic labeling pipeline, for weakly-supervised setting, you still rely on region-level captioning technique to provide the region-level sentence. **So how do we compare this two pipelines' error?** If they have similar annotation error, I don't see the necessity to focus on weakly-supervised setting.


2. **Overclaim about the shortage of previous structures**: In line 74 to lin 76, the authors claim that transformer-based method is less efficient in training than their combination of CNN + transformer. **However, current transformer-based methods do have CNN structure**, e.g. TransVG, QRNet, VLTVG, VG-LAW, SegVG. They all adopt the encoder of DETR which is a ResNet + Transformer Layer. Therefore, I don't see the main technical difference in the structure. Moreover, **the paper didn't provide results to support the claim of more efficient like training cost**.


3. **Similar Grad-CAM method already exists**: <Improving Visual Grounding by Encouraging Consistent Gradient-based Explanations> **has already explore the use of Grad-CAM as a supervision signal**, therefore, the novelty of the main contribution of this paper is limited.

4. **Technical concern about Grad-CAM**: Grad-CAM is not a state-of-the-art post-hoc explanation method given that the authors have involve transformer structure. Related work like <Token Transformation Matters: Towards Faithful Post-hoc Explanation for Vision Transformer> has a much better explanation result. **Therefore, it might be meaningless to supervise Grad-CAM if it is not the actual way of model's decision making.**

5. **Writing needs to improve**: The figure in page 5 is unclear. The tables in page 7 are unclear.

**Questions:**

1. How is the performance when using a different post-hoc explanation method to supervise?

---

> ### Comment · Reviewer_2Uie · 2024-11-26
> **Maintain my reject rate**
>
> Since the authors haven't replied my concerns, I finally decide to maintain my reject rate.

---

### Official Review · Reviewer_K9e7 · 2024-11-02

**Soundness:** 2
**Presentation:** 2
**Contribution:** 2
**Rating:** 3
**Confidence:** 4

**Summary:**

This work focuses on weakly supervised visual grounding task, where only image and language inputs are available without precise object-level annotations. There are three main components In the proposed framework, including an efficient visual encoder that combining the CNN and transformer structures, a CAM encoder that extracting useful information from Grad-CAM features, and a multi-modality fusion module that aggregating visual, language and CAM features. Moreover, an attention mining loss is introduced to make the Grad-CAM features to highlight the whole object instead of focusing on its local parts.

**Strengths:**

1.The architecture of each module is clearly stated, it is not hard to understand their designs.

2.The experiments are comprehensive, which are conducted on five datasets including RefCOCO, RefCOCO+, RefCOCOg, ReferItGame, and Flickr30K. The ablation studies also demonstrate the effectiveness of the proposed modules.

**Weaknesses:**

1. In the related work section, some recently proposed approaches are not appropriately mentioned, such VLTVG [1], QRNet [2], LUNA [3], VG-LAW[4] for fully supervised visual grounding, and CPL [5], WSVG[6], AMC [7], enhanced X-VLM [8] for weakly supervised visual grounding.

2. The authors claim that “no previous attempts have been made to integrate Grad-CAM with existing weakly supervised visual grounding methods”, however, Grad-CAM has already been used in [7] and [8],  could the authors clarify how their use of Grad-CAM differs from or improves upon the approaches in [7] and [8]?

3. Since the introduced attention mining loss was firstly proposed in GAIN [9] method, could the authors clarify their loss differs from that in GAIN or the modifications or improvements they've made? This will help to clarify their contribution.

4. For the experiments, the comparison methods seem to be a bit outdated. All of them are published before 2023, which may not fully demonstrate the superiority and effectiveness of the proposed method. Could the authors compare the proposed method with recent works [5-8], especially with [7] and [8] that also utilize the Grad-CAM technique.

5. It may be better to visualize the extracted CAM features to show their effectiveness on highlighting the target object areas.

6. The texts in Figure 4 are too small and hard to read.

[1] Yang L, Xu Y, Yuan C, et al. Improving visual grounding with visual-linguistic verification and iterative reasoning[C]//Proceedings of the IEEE/CVF Conference on Computer Vision and Pattern Recognition. 2022: 9499-9508.

[2] Ye J, Tian J, Yan M, et al. Shifting more attention to visual backbone: Query-modulated refinement networks for end-to-end visual grounding[C]//Proceedings of the IEEE/CVF Conference on Computer Vision and Pattern Recognition. 2022: 15502-15512.

[3] Liang Y, Yang Z, Tang Y, et al. LUNA: Language as Continuing Anchors for Referring Expression Comprehension[C]//Proceedings of the 31st ACM International Conference on Multimedia. 2023: 5174-5184.

[4] Su W, Miao P, Dou H, et al. Language adaptive weight generation for multi-task visual grounding[C]//Proceedings of the IEEE/CVF conference on computer vision and pattern recognition. 2023: 10857-10866.

[5] Liu Y, Zhang J, Chen Q, et al. Confidence-aware Pseudo-label Learning for Weakly Supervised Visual Grounding[C]//Proceedings of the IEEE/CVF International Conference on Computer Vision. 2023: 2828-2838.

[6] Zhang R, Wang C, Liu C L. Cycle-consistent weakly supervised visual grounding with individual and contextual representations[J]. IEEE Transactions on Image Processing, 2023.

[7] Yang Z, Kafle K, Dernoncourt F, et al. Improving visual grounding by encouraging consistent gradient-based explanations[C]//Proceedings of the IEEE/CVF Conference on Computer Vision and Pattern Recognition. 2023: 19165-19174.

[8] Pham V Q, Mishima N. Focusing on Targets for Improving Weakly Supervised Visual Grounding[C]//ICASSP 2023-2023 IEEE International Conference on Acoustics, Speech and Signal Processing (ICASSP). IEEE, 2023: 1-5.

[9] Li K, Wu Z, Peng K C, et al. Tell me where to look: Guided attention inference network[C]//Proceedings of the IEEE conference on computer vision and pattern recognition. 2018: 9215-9223.

**Questions:**

1. For the visual encoder, I am wondering why combining the transformer and CNN architectures can “reduce computation costs and accelerate the training processing”?

2. For the extraction of CAM features, the authors first need to identify all the nouns in the input referring text. I am wondering that how to avoid or mitigate the effect of non-target object nouns on the extracted CAM features. For example, for the input text “a white dog lying on the sofa”, since “sofa” is not a part of the target object “dog”, the image regions relevant with “sofa” should not be focused by CAM features, so how to reduce its effects?

---

> ### Comment · Reviewer_K9e7 · 2024-11-27
>
> The authors have not replied my comments, so I maintain my rating as "Reject".

---

### Official Review · Reviewer_vmg3 · 2024-11-03

**Soundness:** 3
**Presentation:** 2
**Contribution:** 2
**Rating:** 3
**Confidence:** 4

**Summary:**

This paper proposes a method to enhance weakly supervised visual grounding by integrating Grad-CAM and a new attention mining loss into the model architecture. The authors introduce a "CAM encoder" that uses Grad-CAM heatmaps to help the model focus on the right objects. An attention mining loss is designed to guide the Grad-CAM features to cover the whole object. The proposed architecture combines CNNs and transformers in the visual encoder and includes a multi-modality fusion module to aggregate visual features, language features, and CAM features. The method achieves state-of-the-art performance on several benchmark datasets.

**Strengths:**

- The paper achieves state-of-the-art performance on 4 out of 5 evaluated datasets

-  The ablation studies demonstrate the effectiveness of the CAM encoder and attention mining loss.

**Weaknesses:**

1. The definition and role of the attention mining loss are not sufficiently explained. While the loss is mentioned (Lam = Sc(I∗)), the paper does not clearly define how it is computed or how it integrates into the training process. Referring to previous work (GAIN) without a detailed explanation may leave readers unclear about this key component. Also it is surprising to not have an exact equation on that loss as it is claimed to be a novel contribution of the paper.


2. The term "Grad-CAM features" is used repeatedly throughout the paper, but Grad-CAM typically produces a single heatmap representing the importance of image regions. Referring to this heatmap as "features" is confusing.

3. The Grad-CAM heatmaps are obtained from a ResNet trained on ImageNet, which by design limits the method's ability to detect classes not present in ImageNet. This restricts the method's applicability to a broader range of objects, despite the language model's capability to handle diverse text queries.

4. The whole subsection 3.2 on the “Language Encoder” could have been summarised in “we use a pretrained BERT encoder”. I don’t understand why the authors detail the tokenization and word embedding when it is exactly the same as BERT.

5. In figure 2 it seems that the “Visual Transformer” is used before the CNN architecture but then in Figure 3 the CNN architecture is used before. Also, the naming is weird as the “Visual Transformer” is actually just of self-attention operation (it is also not clear if this module used several attention heads). The authors also say, “We use a self-attention module to capture the local information of the given features”, but in general, self-attention modules are used to capture global information. Hence, the authors need to add more details on that.

6. The paper mentions the use of a "self-taught regression loss" and a "phrase reconstruction loss" following RIF, but does not provide explanations or formulations for these losses. Including details or referring to supplementary materials would enhance clarity.

7. Given that the method is centered around using Grad-CAM, it would be beneficial to include visualizations of the Grad-CAM heatmaps. This would help readers understand how the attention mining loss influences the attention maps and contributes to improved localization.


Miscellaneous: Tables 1 & 2 are quite small, which can be acceptable if the authors lack space, but the paper is only 9 pages while the page limit is 10.

**Questions:**

1.  Can the authors provide a more detailed formula and explanation of the attention mining loss? Specifically, how is Lam computed, and how does it influence the training process to ensure the Grad-CAM features focus on the whole object?

2. How does using a ResNet trained on ImageNet to generate Grad-CAM heatmaps affect the generalizability of the method? Can the model handle objects not included in the ImageNet classes? If not, how might this limitation be addressed?

3. There appears to be a discrepancy between the figures and the description of the visual encoder. Could the authors clarify the sequence of the CNN and transformer modules in the visual encoder? Additionally, how does the self-attention module capture local information, and does it use multiple attention heads?

4. Can you include visualizations of the Grad-CAM heatmaps before and after applying the attention mining loss? This would help illustrate how the loss function improves the focus of the attention maps on the entire object.

---

### Official Review · Reviewer_CwTX · 2024-11-04

**Soundness:** 1
**Presentation:** 2
**Contribution:** 1
**Rating:** 3
**Confidence:** 5

**Summary:**

In this paper, the author propose a weakly supervised visual grounding architecture that combines the transformer and CNN architectures. Observing that Grad-CAM is useful in weakly supervised training, they design a CAM encoder that utilizes the Grad-CAM to provide better object localization when predicting the final bounding box.

**Strengths:**

- The structure of the paper is complete

**Weaknesses:**

- Q1. The first innovation claimed in this paper is the use of Grad-CAM to enhance weakly supervised grounding ability. However, Grad-CAM has been proposed as an attention tool for many years and has been widely utilized in various fields. This paper argues that the utilization of Grad-CAM cannot be considered as an innovation.

- Q2.  The second innovation claimed in this paper is the incorporation of multi-layer features and transformer networks. However, these practices are already widely used in existing grounding systems such as Pseudo-q, CLIP-VG, etc.

- Q3. It is worth saying that Pseudo-q is an unsupervised method. However, it is treated as weakly supervised method in this paper.

- Q4. The work presented in this paper is simple and direct, and its overall innovation is appears relatively low and is far from to meet the standard of top-tier conference paper, especially ICLR. Therefore, it would be advisable for the authors to consider submitting their work to lower-level journals or conferences.

**Questions:**

See weakness.

**Details Of Ethics Concerns:**

No Ethics Concerns

---

### Meta-Review · Area_Chair_hbvV · 2024-12-16

**Metareview:**

This paper proposes a method to enhance weakly supervised visual grounding by integrating Grad-CAM and a new attention mining loss into the model architecture. The proposed method achieves state-of-the-art performance on several benchmark datasets. All reviewers are concerned about the limited novelty and unclear clarifications in this paper. They recommend rejection and the authors did not participate in the rebuttal. So the final decision is reject.

**Additional Comments On Reviewer Discussion:**

No response from the authors.

---

### Decision · Program_Chairs · 2025-01-22

Reject